# HESS Opinions: Science in today's media landscape — challenges and lessons from hydrologists and journalists

Stefanie R. Lutz[1], Andrea Popp[2,3,4], Tim van Emmerik[4,5,6], Tom Gleeson[7], Liz Kalaugher[8], Karsten Möbius[9], Tonie Mudde[10], Brett Walton[11], Rolf Hut[5], Hubert Savenije[5], Louise J. Slater[12], Anna Solcerova[5], Cathelijne Stoof[13], Matthias Zink[14]

[1] Department of Catchment Hydrology, UFZ Helmholtz Centre for Environmental Research, Theodor-Lieser-Str. 4, 06120 Halle, Germany

[2] Department Water Resources & Drinking Water, Eawag, Überlandstrasse 133, 8600 Dübendorf, Switzerland

[3] Department of Environmental Systems Science, ETH Zurich, Rämistrasse 101, 8092 Zürich, Switzerland

[4] Young Hydrologic Society

[5] Water Resources Section, Delft University of Technology, Stevinweg 1, 2628 CN Delft, The Netherlands

[6] Amsterdam Institute for Metropolitan Solutions, Mauritskade 62, 1092 AD Amsterdam, The Netherlands

[7] Department of Civil Engineering and School of Earth and Ocean Sciences, University of Victoria, 3800 Finnerty Road, Victoria BC V8P 5C2, Canada

[8] environmentalresearchweb, IOP Publishing, Bristol, UK

[9] Mitteldeutscher Rundfunk, Leipzig, Germany

[10] De Volkskrant, Amsterdam, The Netherlands

[11] Circle of Blue, Traverse City, USA

[12] Department of Geography, Loughborough University, Loughborough, LE11 3TU, UK

[13] Soil Geography and Landscape Group, Wageningen University and Research, Droevendaalsesteeg 3, 6708 PB Wageningen, The Netherlands

[14] Department Computational Hydrosystems, UFZ Helmholtz Centre for Environmental Research, Permoserstraße 15, 04318 Leipzig, Germany

*Correspondence to*: Stefanie R. Lutz (stefanie.lutz@ufz.de)

**Abstract.** Media such as television, newspapers and social media play a key role in the communication between scientists and the general public. Communicating your science via the media can be positive and rewarding by providing the inherent joy of sharing your knowledge with a broader audience, promoting science as a fundamental part of culture and society, impacting decision and policy makers, and giving you a greater recognition by institutions, colleagues and funders. However, the interaction between scientists and journalists is not always straightforward. For instance, scientists may not always be able to translate their work into a compelling story, and journalists may sometimes misinterpret scientific output. In this paper, we present insights from hydrologists and journalists discussing the advantages and benefits as well as the potential pitfalls and aftermath of science-media interaction. As we perceive interacting with the media as a rewarding and essential part of our work, we aim to encourage scientists to participate in the diverse and evolving media landscape. With this paper, we call on the scientific community to support scientists who actively contribute to a fruitful science-media relationship.

**1 Why interact with today's media landscape?**

In this partisan era filled with 'alternative facts', it is essential for science and scientists to be transparent and communicative to the general public (Kirchner, 2017). Presenting scientific methods and the work of scientists in general can contribute to
people's understanding of the scientific pursuit of facts and reduce scepticism towards science (e.g., regarding climate change or vaccinations; Hamilton et al., 2015). For many scientists, the main objectives behind engaging with the public are to inform and educate, oppose public misinformation and generate excitement about science (Dudo and Besley, 2016). Science communication may also combat the prevalent stereotype of the old, white and male scientist sitting in an ivory tower that the media have been inclined to show (Hut et al., 2016) and thereby inspire children and minority groups to
pursue a career in science.

Interacting with the media is one aspect of science communication that can be highly rewarding for scientists and comes with numerous benefits (Fig. 1). For example, it can improve public education and attitude towards science, contribute to policy making and public debate, stimulate acknowledgement as well as critical reflection of scientific work, and increase the recognition of scientists (Dijkstra et al., 2015; Peters et al., 2008). Accordingly, despite continuous scepticism towards
media in general, most scientists describe their personal interactions with journalists as positive (Besley and Nisbet, 2013; Peters et al., 2008). From our own experience, interacting with the media brings the inherent joy of being able to communicate research findings to the broader public, thereby promoting science as a fundamental part of society (Fig. 1). Moreover, science journalism can result in your work having more impact on decision and policy makers, extend your network among non-academics and give you a greater recognition by your institution, colleagues and funding agencies,
which also increases your chances to obtain grants.

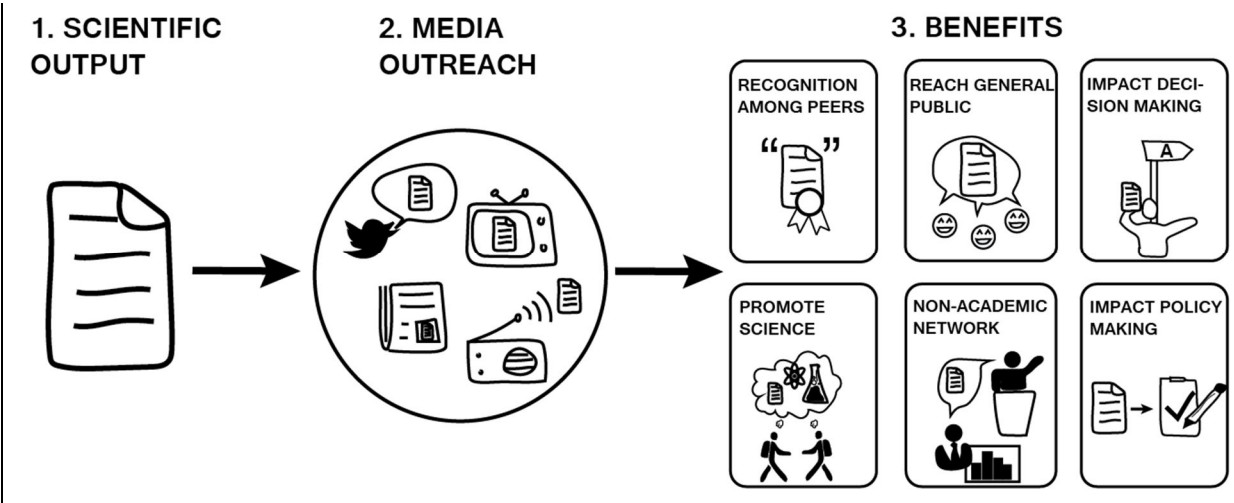

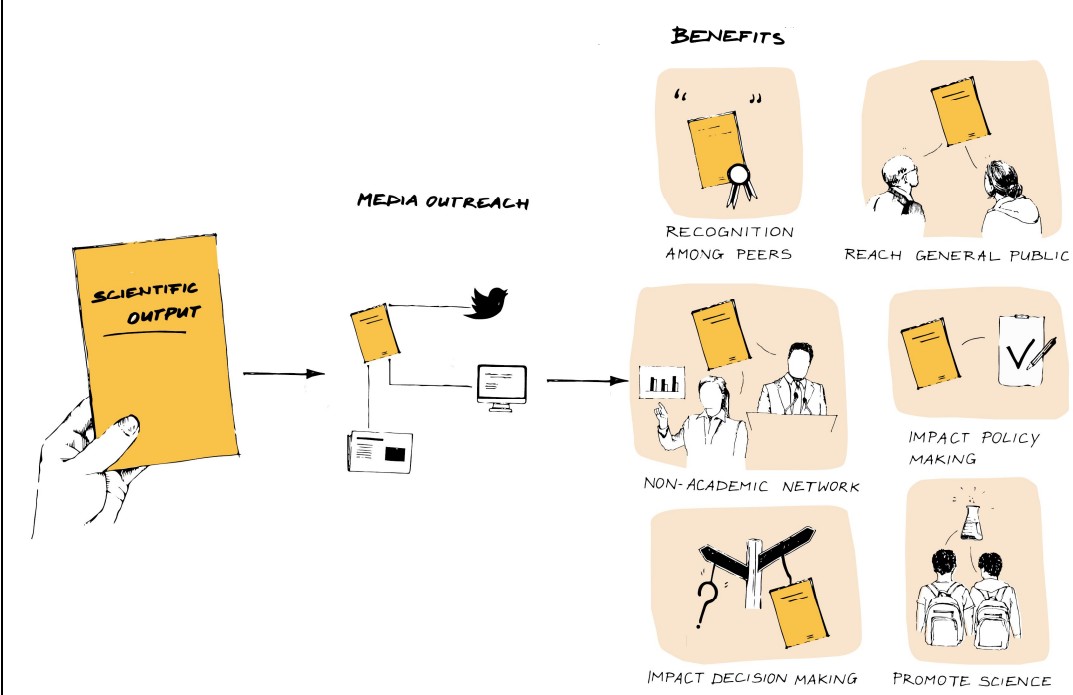

**Figure 1: The benefits of communicating your research via the media.**

A recent study suggests that nearly 18% of natural science papers (published between 1980 and 2012) remain uncited and thus go unnoticed by the scientific community (Van Noorden, 2017). Although this is based on citation databases with known issues (e.g., also counting publications such as book reviews, commentaries and errata, which are not intended to be cited), it does illustrate how many natural science papers get little attention by scientific peers and the general public. Correspondingly, a review of two major US media outlets has shown that while the number of peer-reviewed articles has considerably increased in recent years, the number of those referenced to in the media remains small (Suleski and Ibaraki, 2010). Hence, we believe that it becomes increasingly important for scientists to acknowledge their 'media responsibility' and to convey their most relevant messages convincingly. At the same time, there is an increasing pressure on scientists to provide newsworthy, controversial or surprising stories (Brown, 2012), and on journalists to provide more scientific stories in less time (Brumfiel, 2009). As a result, inaccuracies in science reporting – albeit moderate and unintentional – can be frequently found even in renowned media outlets (Vestergård, 2011; Singer, 1990). Similarly, scientists are not immune from drawing misleading or premature conclusions in order to increase the perceived relevance of their findings (Chiu et al., 2017).

Traditionally, science journalism has been understood by many scientists as a unidirectional process to inform and increase public understanding (Nielsen et al., 2007), largely controlled by a few journalistic gatekeepers that filter and process the original information for the public (Mazur, 1981). Consequently, many scientists share their findings with the media only once they have been published in a scientific journal (Peters, 2013). Some scientists also perceive dealing with the media as

a delicate task that can lead to improper quotations or misrepresentations of research results (Dijkstra et al., 2015; Stewart and Nield, 2013) and decrease their recognition among colleagues (The Royal Society, 2006; Willems, 2003). Concurrently, many journalists describe difficulties in finding interviewees who are willing and able to speak on pressing topics (Dijkstra

et al., 2015). Nonetheless, science-media interaction has generally increased in recent years, in part because science communication is progressively being considered integral to a scientist's occupation (Dijkstra et al., 2015; Peters, 2013; Tsfati et al., 2011). Moreover, some funding bodies require grant proposals to specify science communication and outreach activities (e.g., obligatory for the EU Horizon 2020 Marie Skłodowska-Curie actions). Hence, there is an essential need to reduce misunderstandings and strengthen the science-media relationship.

Scientific knowledge is increasingly consumed online, via blogs, social networks or news aggregators, which provide multimedia content and tools for interaction with other users (Brossard, 2013; Peters, 2013). These online sources offer the opportunity to rapidly access and share information among scientific peers and with the public in an open and participatory environment (Collins et al., 2016; Watermeyer, 2010). Compared to traditional media, this new way of sharing information may, however, complicate the distinction between scientific results, opinions and user comments, while presentation type,

format and user comments become more important for the perception of scientific content (Brossard, 2013). From the science journalist's perspective, the rise of online media has replaced the journalists' main function as science translator and gatekeeper with more participatory and interactive roles such as public intellectual and educator as well as 'curator' of scientific information (Fahy and Nisbet, 2011).

In light of the benefits of efficient science-media interaction, the aim of this commentary is to encourage scientists to

participate in today's diverse media landscape. To facilitate this, we discuss the advantages and benefits as well as the potential pitfalls and aftermath of media interaction for scientists, with a focus on geosciences and hydrology. In order to reflect both perspectives of science-media communication, we also include the opinion of four journalists from different media outlets (i.e., newspaper, online media and radio). With this commentary, we do not seek to provide a comprehensive review of the science-media relationship, but rather discuss the importance of strengthening the relationship between

scientists and journalists, and provide concrete suggestions based on input from both perspectives. While applicable to other scientific fields, this paper is particularly aimed at hydrologists and geoscientists.

In section 2, we highlight four examples in which media coverage of scientists had an unforeseen or unwanted outcome. Although we perceive working with journalists generally as a positive experience, we focus on four challenging examples to highlight the pitfalls and help other scientists avoid similar situations. In section 3, we summarize the lessons learned from

the four examples and give some general advice on science-media interactions from a scientist's point of view. Section 4 examines science-media interaction from the perspective of (science) journalists and the underlying principles of science journalism. The commentary concludes with a synthesis of the discussion and an outlook on how to strengthen the science-media relationship.

## 2 The challenges of communicating science via the media

Dozens of papers and books have been written on effective science communication with the media and the public (e.g., Cooke et al., 2017; Illingworth and Allen 2016; Bubela et al., 2009; Weigold, 2001), yet it still remains a challenge for all parties involved (National Academies of Sciences, Engineering, and Medicine, 2017). Since anecdotes can be effective representations of broader trends (Berg and Seber, 2016), we provide several first-hand examples of how geoscientists experienced the challenges of science-media interaction, despite good intentions and preparation.

*Flood example: exaggeration can lead to false conclusions drawn by the media*

Exaggeration of scientific claims can draw media attention, but can also 'go wrong'. In the mid-nineties, the Netherlands experienced major flooding, and Professor Hubert Savenije (Delft University of Technology) was called upon to discuss this disaster. Contrary to Professor Savenije's expectations, the interview, containing minor exaggerations (such as his comment that 'for the Dutch Ministry, the Meuse river starts at the border'), resulted in a front page article suggesting that the water

authorities had a poor understanding of Dutch rivers (De Volkskrant, 1995). The ministry responsible for flood management was highly offended, and the story was repeatedly featured on the news for several days through various media outlets. In the end, two follow-up articles in longer-format outlets gave Professor Savenije the opportunity to provide a more representative and nuanced perspective (Savenije, 1995a, b).

*Fire example: unintentional early releases of sensitive topics can result in criticism and bias*

During preparation of an opinion paper on ecological effects of wild and traditionally managed fires on UK peatlands (Davies et al., 2016a), the authors planned a press release to take the lead in the communication of this paper. Fire management is a highly political and emotive topic in the UK, making it crucial to control potential media attention. Due to new regulations in the UK, designed to satisfy the UK's Research Excellence Framework guidelines, the accepted manuscript was made publicly available upon acceptance through a university repository. This was a result of misunderstandings

between the authors and the scientific journal about embargo terms for the repository. Ironically, the paper that called for informed, unbiased debate was then misrepresented and taken out of context by groups with divergent environmental, social and political agendas. This led to significant criticism from some commentators who claimed that the paper had been leaked to an organization on the opposing 'side' of the debate, which in turn was used as a pretext to accuse the authors of bias and to call their credibility into question (cf. Davies et al., 2016b).

*Drought example: journalists might seek after provoking statements*

User-friendly maps can be a valuable information tool for the media and the public. For example, the German Drought Monitor (Zink et al., 2016) presents near real-time, online soil moisture information in illustrative maps of daily soil drought conditions. As a consequence, the German Drought Monitor is frequently used by several regional and national newspapers

as well as television stations to inform the public about the recent status of soil moisture conditions during drought events.

Due to its large influence, the scientists who had developed the German Drought Monitor were frequently approached by journalists during the 2015 drought in Germany. Some of these journalists tried to prompt the scientists to state that this drought was 'the worst drought ever recorded', or that this drought could 'directly be related to climate change', although the scientists were not able to draw such general conclusions from their results at that stage of the event.

*Groundwater example: journalists can distort results by taking them out of context*

During a press conference at the European Geosciences Union (EGU) General Assembly 2017, Professor James Kirchner (ETH Zurich) reported on a recent paper that he co-authored (Jasechko et al., 2017). This paper stated that fossil groundwater can contain a small fraction of water less than 50 years old, as evidenced by detectable levels of tritium remaining from nuclear bomb testing in the 1950s. The authors concluded from this tritium signal that even fossil groundwater can potentially contain some percentage of much more recent water, and thus be vulnerable to modern

contamination. Subsequently, the Daily Mail (a British tabloid) published an article that took the statements given by Professor Kirchner out of context, complete with the headline 'Groundwater drunk by BILLIONS of people may be contaminated by radioactive material spread across the world by nuclear testing in the 1950s' and a stock photo of a mushroom cloud (The Daily Mail, 2017). This article further stated that tritium in drinking water is linked to an 'increased risk of mutations and cancer', suggesting that groundwater might be harmful to consume. The Daily Mail article greatly

misled the public by taking statements out of context, exaggerating them and drawing false conclusions. Such articles can produce a general mistrust in public water supply, ignoring the fact that drinking water in developed countries is strictly regulated and extensively monitored. The Daily Mail ignored the request by Kirchner and Jasechko that the story be corrected or retracted (Kirchner, personal communication).

These four examples highlight some of the challenges in the communication between scientists and journalists that may arise

from exaggeration of scientific results, dealing with controversial topics, the risk of miscommunication, the difficulty of communicating uncertainty in research results, and misrepresentation and improper quotation of studies. These challenges may make engaging with the media feel like trying to cross a large divide on a wiggly bridge (Fig. 2).

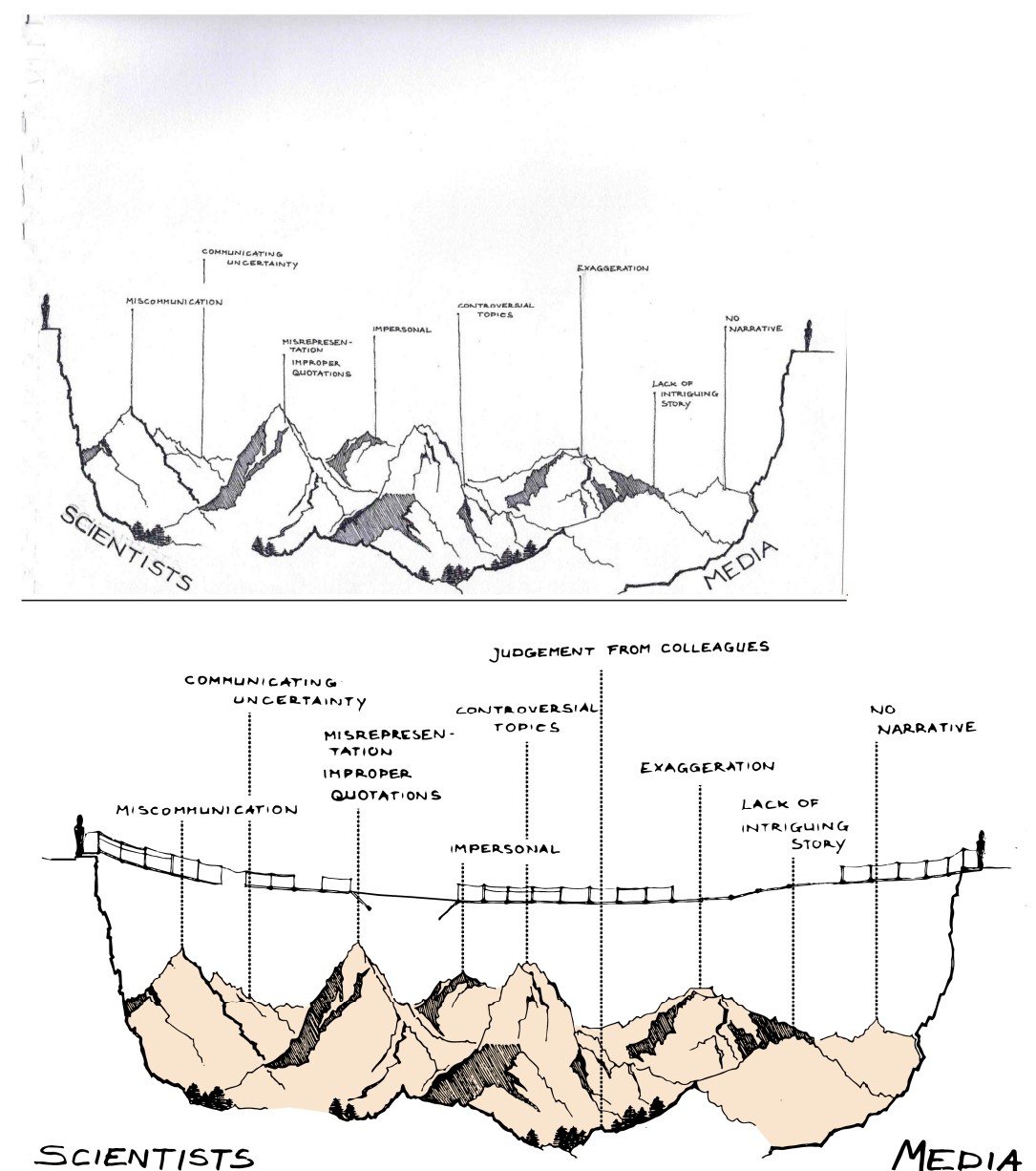

**Figure 2: Challenges and fears of scientists and journalists in science communication.**

## 3 Science reporting from a scientist's point of view

The flood example reveals that exaggeration can help to draw media attention but also lead to major miscommunications. Scientists should thus always be careful when exaggerating or using strong language. In the case of papers dealing with sensitive and controversial topics such as the fire example, authors should ensure that embargo terms are strictly enforced in

repository depositions to prevent any preliminary release of findings. It should be noted that, along with the negative media coverage, the paper in the fire example was well received by many working directly in the field of fire ecology and by land managers from organizations that are on opposing 'sides' in the debate. In the drought example, the researchers used the opportunity to give insights into drought mechanisms and the quantification and benchmarking of drought events, instead of agreeing with the journalists' suggestion that the current event could be directly attributed to climate change. Moreover, the

communication between the scientists and journalists improved once the scientists refrained from using expert terminology (e.g., precipitation instead of rain) and provided comprehensible examples to explain the implications of their findings. This suggests that it can be advisable for scientists to reflect on the detail of information they would like to communicate to their interviewer in order to avoid misunderstandings about their research. Hence, scientists may wish to inquire about the journalist's background before answering specific questions, and should be cautious when communicating uncertain

conclusions from their research results. Finally, the groundwater example shows that even with all possible advice taken into account, some media outlets might decide to explore an angle that is not there and scientists will not be able to entirely prevent distorted media coverage of their research.

Complete control over communication and media attention by scientists is unrealistic and undesirable as we need critical and independent media to challenge the validity of scientific studies. In addition, refuting incorrect stories does not necessarily

decrease misperceptions and can even lead to a larger public belief in the misleading or incorrect story (Lewandowsky et al., 2012; Nyhan and Reifler, 2010). Hence, in this case, possibly the best strategy for scientists is to provide accurate and truthful contributions, and to accept that misleading reporting such as in the groundwater example can happen. Fortunately, our experience is that the majority of media coverage is reasonable and nuanced, as also illustrated by other articles on the groundwater example (see, e.g., Amos, 2017).

While journalists may not be willing to send their writing before publication, inaccuracies or faulty conclusions may be avoided if scientists ask journalists to allow verification of direct quotes or discussion of crucial statements from interviews. Similarly, when issuing a press release, scientists can try to liaise closely with press officers to ensure a balanced and accurate press release. If this is not possible – for example, due to rigid deadlines – and a press release or journalistic report ends up containing errors, scientists can suggest a polite correction to the journalist or press officer.

In addition to avoiding the pitfalls illustrated here, there are also numerous ways how to actively improve science-media communication. First, we suggest that establishing a digital presence is key to increase your visibility and accessibility as a scientist – both in the media and among peers. Second, you can strengthen the clarity and comprehensibility of your work by distilling your key messages in two or three concise messages and by using real-life examples. We believe that research findings can be explained more easily if scientists present them in natural language, use catchy titles and show why the

public should care about the line of research and science in general. Thirdly, you might want to include pictures and personal details in your work and communication in order to make your story unique and help people remember you and your research. In addition, you can add a personal note by not only reporting the scientific facts, but also describing any exciting events or challenges that occurred during your research. It can be most effective to adapt a style of narrative storytelling,

where not only the base facts are important but also the plot, so that drama and tension will keep the audience engaged with the topic (Hut et al., 2016).

## 4 Science reporting from a journalist's point of view

News media, especially in the fast pace of the internet age, are driven by what is new. This implies that researchers should expect strong initial interest in a study and a sharp decline in the days following its release. It is crucial that scientists are available for interviews in that high-interest period. However, especially with social media, science communication does not end after a few days following a study's publication in the media. Social media allows for continued engagement, both with reporters and the general public. It can be scary terrain for scientists, but the outcomes – exposure and helping to guide the dialogue – can be highly beneficial. For example, Twitter messages or blog posts from scientists can help journalists develop a relationship with scientists that results in stories about their research. Social media also enables journalists to learn about studies, research interests, and the research questions that scientists are most excited about. The use of blogs and social media can, in turn, help scientists improve their communication skills outside the realm of scientific journals.

Besides novelty, the main factors influencing news coverage are narrative, conflict, and familiarity. This means that news organizations do not simply repeat information. Instead, they select from the abundance of news those items that match their worldviews, interests, or capabilities, and thus establish narratives and context around news. Scientific stories themselves are also narratives, and the easiest form of narrative is the conflict narrative, i.e., side A vs. side B. or new idea vs. existing policy. In particular, research that fits existing conflict narratives or is familiar to the reader is more likely to be picked up (see also Downs, 2014; Stewart and Nield, 2013). For example, a narrative might arise from conflicts between local residents of a flood-prone area who favour the reinforcement of floodwalls and embankments along the river, and advocators of a more "natural" flood management who advise using the residential area as the natural floodplain of the river. Another example of a conflict narrative that has been frequently used (while being simplistic) is the "farmer vs. fish" narrative, which refers to water use restrictions for farmers to alleviate the pressure on natural water resources during severe droughts in California (e.g., Kloberdanz, 2008). Therefore, as narratives will help conveying your message, prepare for an interview with a journalist by determining the unique points, societal relevance and narrative thread of your research story. The more enthusiastic you are about your topic, the easier it will be for journalists to convey this enthusiasm to the public.

Journalists have to ensure that their article has a clear, striking message that will grab the reader's attention. Otherwise, the article might not be published or the message may become so weak that the reader will not read the entire article. For journalists, style is just as important as content, whereas for academic publications, content takes priority and style is defined by academic writing standards. Therefore, be well prepared to present clear results that back up a strong message before contacting a journalist. Good narratives or storylines can serve as discussion starter and facilitate communication between you and the journalist during the interview.

Journalists often ask other researchers to comment on a study in order to obtain an independent second opinion. This is standard practice for good science journalism and helps journalists better assess the novelty and impact of the research findings and assure themselves of their validity. For example, second opinions were highly valuable in the context of a study that reported a substantial increase in break rates of water pipes in recent years (Folkman, 2018). After scientists who were asked for a second opinion had raised concerns about the scientific methods, the journalist refrained from his plan to report on the study. Journalists value scientists who are not afraid to discuss uncertainty, and who are forthright about any assumptions their research is based on. As scientists are accustomed to collaborative writing and peer review, they sometimes offer to review quotes or the entire story before publication. However, journalists, valuing independence, are generally reluctant to send quotes, let alone the entire story. Depending on organization policies or personal preference, they might only send the parts of the article that quote the scientist directly. Establishing trust between journalists and researchers is particularly important in this regard.

One of the biggest obstacles to effective communication with the media might be scientific training itself. As historian Naomi Oreskes emphasized at the American Geophysical Union (AGU) Fall Meeting 2016, the key to good communication is keeping the message simple and telling a memorable story by mentioning something personal or evoking emotions (Kalaugher, 2016). However, to many scientists, 'simple feels simplistic' and stories feel made up, according to Oreskes. She believes that scientists often think that research should be impersonal, unemotional and dispassionate, whereas scientific studies have actually shown that emotion is an essential part of reasoning (e.g., Kahan 2010, 2015). Consequently, scientists may tend to provide stories lacking personal anecdotes and intriguing narratives, which can further complicate the communication between journalists and scientists (Fig. 2). Therefore, as a scientist, do not be afraid of including emotional aspects by, for instance, showing how scientific findings could affect people's lives. At conferences and other science-media events, you can seize the opportunity to approach journalists after your presentation to ask whether the societal implications of your findings are clear, particularly if these could not be addressed in detail due to the scientific nature of the presentation. In general, you can and should practice talking about your research to friends, relatives or strangers, which gives you instant feedback on the aspects that are most interesting to non-scientists.

**5 Strengthening the science-media relationship**

We believe that both scientists and journalists have a duty to enhance and strengthen the current science-media relationship. Therefore, it is essential to understand which aspects in this relationship are most important for scientists and journalists. We propose the following to facilitate the dialogue between science and the media. First, we as scientists should be well informed about the type of audience for which a journalist reports. For example, if you are interviewed for a magazine read by technical experts, keep in mind that this magazine requires a different content and style than a short news brief. Second, while scientists tend to look at things from their own area of expertise, it is essential for good science reporting to zoom out and look at the bigger picture. If you fail to put your research into perspective, it will be the journalist's responsibility to do

so – which can increase the risk of distortions or faulty conclusions. Finally, be prepared to discuss the backstory of your research findings. Reporters ask for narratives and like to personalize your research by highlighting, for example, what drives your interest in the topic and what obstacles were encountered during your research project. These suggestions may serve as a basic recipe for successful communication with journalists (Fig. 3), which you can amend according to your taste and the specific conditions of your interaction with the media.

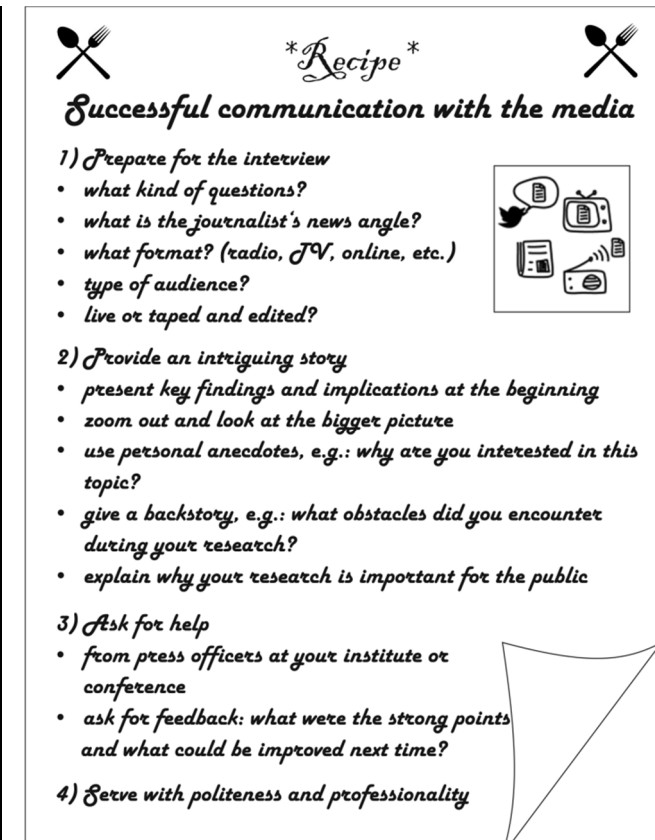

*Recipe*

**Successful communication with the media**

1) Prepare for the interview
- what kind of questions?
- what is the journalist's news angle?
- what format? (radio, TV, online, etc.)
- type of audience?
- live or taped and edited?

2) Provide an intriguing story
- present key findings and implications at the beginning
- zoom out and look at the bigger picture
- use personal anecdotes, e.g.: why are you interested in this topic?
- give a backstory, e.g.: what obstacles did you encounter during your research?
- explain why your research is important for the public

3) Ask for help
- from press officers at your institute or conference
- ask for feedback: what were the strong points and what could be improved next time?

4) Serve with politeness and professionality

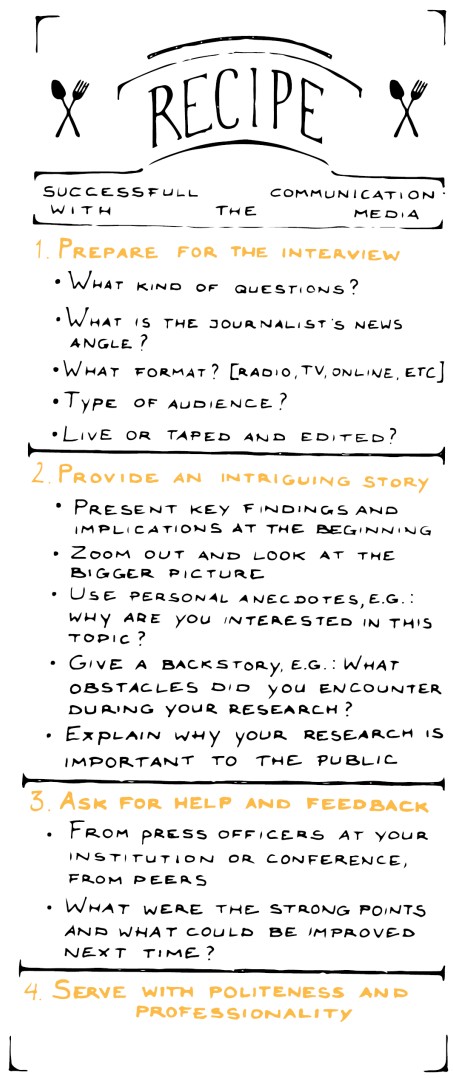

**Figure 3: Our suggestions for successful communication between science and the media.**

As the media world may be uncharted waters for many scientists, our geoscience community needs to continuously encourage scientists to get involved in the media landscape and actively contribute to a better science-media relationship. We should support and reward scientists that communicate research to the media and the public. One way to do so might be the use of 'media altmetrics' that measure a scientist's engagement with the media and public, similarly to existing metrics that count the number of news articles a paper has featured in (cf., Priem, 2013). We endorse initiatives that aim for formal

recognition of science communication as an important scientific activity besides teaching and research (e.g., Rathenau Instituut, 2017).

Communication skills can be practiced and developed in science communication training, where scientists are provided with the tools they need to effectively communicate with the media. Hence, we would like to stress the importance of media

training for scientists in their early career stage through their institutes or organizations. We strongly encourage scientists engaging with the media to seek advice at their institutional press office or from other professional resources such as the Science Media Centre (http://www.sciencemediacentre.org/), and to inquire about science communication courses offered by graduate schools, universities or funding bodies (e.g., courses for grant holders of the European Research Council or the UK's Natural Environment Research Council). We also propose organisation of joint media training workshops and informal networking sessions for both scientists and journalists, which has already become part of the programme of the EGU General Assembly in recent years (e.g., the short course 'Communicating geoscience to the media'; Ferreira et al., 2018). Large geoscience conferences such as the EGU General Assembly or the AGU Fall Meeting are suitable platforms for such workshops. Moreover, while most discussions in media rooms revolve around the latest studies, conferences and science-media networking events are also the place to develop longer-term relationships between scientists and journalists. In summary, the pillars that we believe support the bridge and facilitate communication between scientists and journalists are an atmosphere of mutual trust, the effort to provide scientific stories with a good narrative and personal aspects, science communication training for scientists and joint media training workshops, assistance by press officers, and support for science communication by the scientific community (Fig. 4).

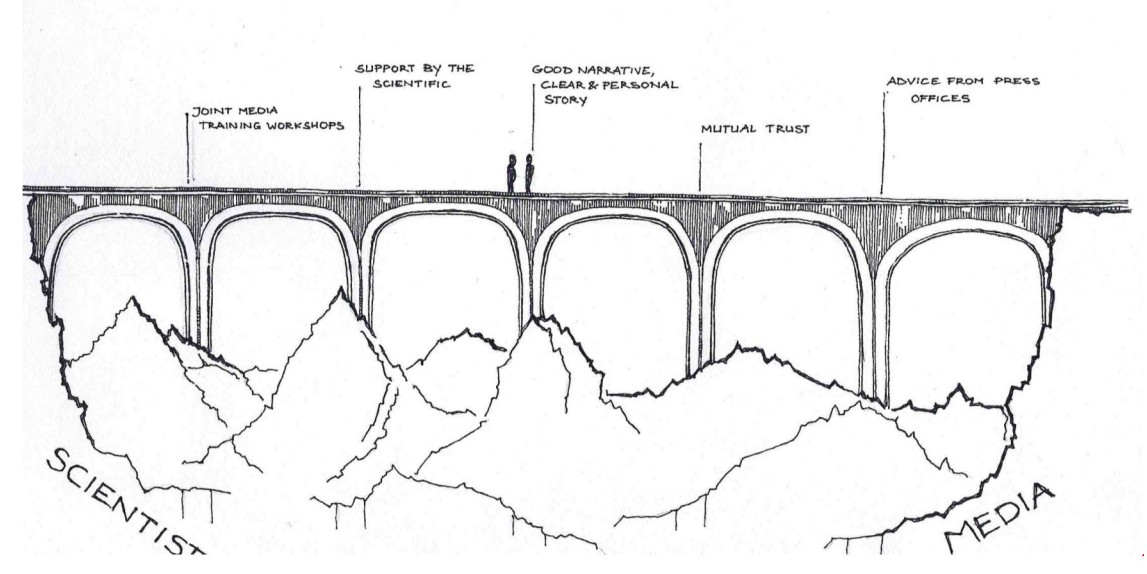

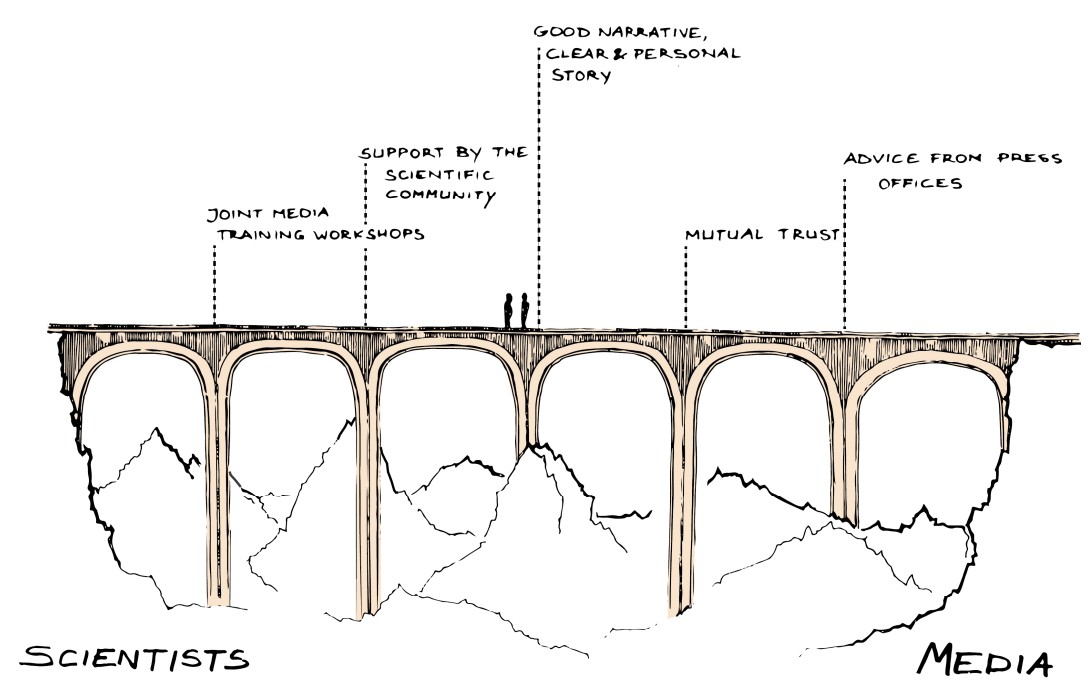

GOOD NARRATIVE, CLEAR & PERSONAL STORY

SUPPORT BY THE SCIENTIFIC COMMUNITY

ADVICE FROM PRESS OFFICES

JOINT MEDIA TRAINING WORKSHOPS

MUTUAL TRUST

SCIENTISTS

MEDIA

**Figure 4: Summary of the tools and success factors for effective science-media communication.**

Reporters are often (and sometimes justifiably) criticized for repeatedly referring to the same scientists for a particular topic. This results both from the journalists' preference for renowned scientists, and from the tendency of public relation departments to favour these scientists over less established ones (Peters, 2013). Hence, to broaden the field of experts that can contribute to the public dialogue, the scientific community needs to promote the voices of underrepresented groups (e.g., women, minority groups and early career scientists) in the conversation. Similarly, universities and science organizations should maintain lists of experts that are available to comment on particular topics and can be approached at scientific conferences. Fortunately, many scientific institutes have such experts available and the American Geophysical Union regularly provides lists of scientific experts, as, for instance, for the 2017 Climate Science Special Report (USGCRP, 2017).

We believe that social media use as well as science communication by university departments and science organizations are crucial aspects in reinforcing the science-media relationship, as they increase and facilitate the interaction between scientists, journalists and the public. Social media in particular offer the opportunity to share information with a broad audience, facilitate networking between journalists and scientists, and foster collaboration and innovation through direct feedback from the public (Hunter, 2016). They are low-threshold means of engagement with the public and offer a more democratic and participatory way of communication compared to traditional media, which may encourage especially young scientists to get involved in science communication. Last but not least, effective science-media interaction also depends on engaged science communication officers who are aware of the current research questions and projects and can, based on that, encourage and

support scientists to participate in science outreach. Hence, we propose further investment in science communication infrastructure to share best practices on how to inform the media and the general public about scientific outcomes.

We hope that the insights and advice shared in this collaborative effort of scientists and journalists will inspire scientists to get involved in science-media communication, and ultimately strengthen the dissemination of scientific results to the public. Both scientists and journalists have emphasized the importance of building narratives around scientific facts and using emotional and personal stories to convey information. This shows that the traditional roles of scientists and journalists in science communication are changing from a unidirectional dissemination of scientific knowledge towards a relation where scientists and journalists can better understand each other's disciplines and work more closely together. We hope that this

commentary will further contribute to a more symbiotic relation between science and the media in today's partisan world.

**Acknowledgements**

This is a joint initiative of geoscientists (S.R.L, A.P., T.v.E., T.G., R.H., H.S., L.J.S., A.S., C.R.S. and M.Z.) and science journalists (L.K., K.M., T.M. and B.W.). Our commentary is inspired by the EGU General Assembly 2017 session 'How my water research made the news', in which hydrologists shared their experiences with communicating science via the media.

We thank James Kirchner, Bárbara Ferreira, Sam Illingworth, Andri Bryner and G. Matt Davies for their valuable comments, which improved this paper significantly. This project was in part funded by the European Union's Horizon 2020 research and innovation programme under the Marie Skłodowska-Curie grant agreements No. 641939 (A.P.) and No. 706428 (C.R.S.). S.R.L. was financially supported by the European Union under the Seventh Framework Programme (Grant agreement no. 603629-ENV-2013-6.2.1-Globaqua).

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
