# Peer review of "HESS Opinions: Science in today's media landscape — challenges and lessons from hydrologists and journalists"

_Hydrology and Earth System Sciences, 2018_

## Editor Comment (EC1) · H.L. Cloke (Editor) · 16 Apr 2018

This is a clear and interesting opinion paper which introduces some important considerations about the interactions of scientists and the media. The authors provide some very interesting examples of pitfalls that we should avoid and some really helpful suggestions of how to improve science media communication.

I wonder if for some of these points to be communicated effectively the authors might follow their own advice a little more and find further visual and engaging materials to highlight their case? Often in science media we see bespoke figures, perhaps blending photos or cartoons or similar to communicate the main points quickly (fig 1 is a good

start). Although an opinion article I see that this particular subject lends itself strongly to several accompanying figures.

Can you use actual examples e.g. of what an actual example of conflict narrative looks like (link?) - this would be much clearer for those who are new to this. The same with second opinions. There is a real chance to provide an important resources here for those considering undertaking this. In a similar way I also think that the 4 examples at the beginning could do with some further details - what is a minor exaggeration? What could you say for the drought example instead of what the journalists want you to say? Can you provide the example of good practice? If the media outlet ignored the request to retract, what on earth can we do as scientists? What damage limitation techniques can we draw on?

As you already point out it can be a scary prospect to start engaging with the media - perhaps you can make an even clearer list of recommendations/steps that someone new can take and point to other resources. I think the role of the press office at institutions, other bodies like the Science Media Centre and Research Council training courses are important and yet not wholly mentioned. Would you agree?

———————————————

---

## Author Comment (AC1) · 15 May 2018

**Reply to editor for manuscript no. hess-2018-13**

Note: Editor comments are shown in italics and author replies are shown in regular font.

References to lines refer to the revised manuscript with tracked changes, provided below the response letter.

**Editor comments:**

*This is a clear and interesting opinion paper which introduces some important considerations about the interactions of scientists and the media. The authors provide some very interesting examples of pitfalls that we should avoid and some really helpful suggestions of how to improve science media communication.*

*I wonder if for some of these points to be communicated effectively the authors might follow their own advice a little more and find further visual and engaging materials to highlight their case? Often in science media we see bespoke figures, perhaps blending photos or cartoons or similar to communicate the main points quickly (fig 1 is a good start). Although an opinion article I see that this particular subject lends itself strongly to several accompanying figures.*

> We thank Hannah Cloke for her positive feedback and highly valuable comments. We agree with the editor that the manuscript would largely benefit from further visuals to emphasize our points. Hence, we added three additional figures (Fig. 2–4; see below). Figure 2 compares the way scientists might perceive communication with the media to "trying to cross a large divide on a wiggly bridge" (line 161). The potential challenges that may arise are illustrated as the sharp rocks and are taken from the feedback of the scientists and journalists contributing to the manuscript. Crossing this divide looks much safer on the solid bridge supported by five pillars in Fig. 4, representing our main suggestions for successful communication between scientists and journalists. Figure 3 gives a more detailed list of recommendations for scientists, which we consider as "basic recipe for successful communication with journalists" (line 268) that can be amended according to the specific situation. The figures are currently sketches but will be replaced by improved high-quality versions for the revised manuscript. In particular, the "wiggly bridge", which is now missing in Fig. 2, will be added to the final version of the figure.

*Can you use actual examples e.g. of what an actual example of conflict narrative looks like (link?) – this would be much clearer for those who are new to this. The same with second opinions. There is a real chance to provide important resources here for those considering undertaking this. In a similar way I also think that the 4 examples at the beginning could do with some further details - what is a minor exaggeration? What could you say for the drought example instead of what the journalists want you to say? Can you provide the example of good practice? If the media outlet ignored the request to retract, what on earth can we do as scientists? What damage limitation techniques can we draw on?*

> We thank the editor for these important suggestions and provided more examples in the revised manuscript. First, we illustrated conflict narratives by the example of a flood-prone area, whose residents "favour the reinforcement of floodwalls and embankments along the river", whereas

"natural" flood management would require "using the residential area as the natural floodplain of the river" (lines 219-221). We also mentioned the "farmer vs. fish" narrative referring "to water use restrictions for farmers to alleviate the pressure on natural water resources during severe droughts in California" (lines 221-224). Second, for the flood example, we provided an example for the "minor exaggerations" from the interview with Prof. Savenije (lines 117-118). Third, for the drought example, we clarified that "the researchers used the opportunity to give insights into drought mechanisms and the quantification and benchmarking of drought events, instead of agreeing with the journalists' suggestion that the current event could be directly attributed to climate change", and underlined the importance of clear language and comprehensible examples (lines 170-174). Finally, we specified in the revision that second opinions are "standard practice for good science journalism" and are the way for journalists to "assure themselves of their [the research findings] validity" (lines 234-235). We aimed at illustrating this by the example of "a study that reported a substantial increase in break rates of water pipes" (line 236), which was called into question by scientists providing a second opinion.

We believe that scientists cannot do much if a media outlet ignored the request to retract other than to "provide accurate and truthful contributions, and to accept that misleading reporting such as in the groundwater example can happen" (see original manuscript). We discourage scientists from engaging in legal battles with media outlets, as this might take up a lot of the scientist's time and energy, especially considering that "refuting incorrect stories does not necessarily decrease misperceptions and can even lead to a larger public belief in the misleading or incorrect story (Lewandowsky et al., 2012; Nyhan and Reifler, 2010)" (see original manuscript).

*As you already point out it can be a scary prospect to start engaging with the media - perhaps you can make an even clearer list of recommendations/steps that someone new can take and point to other resources. I think the role of the press office at institutions, other bodies like the Science Media Centre and Research Council training courses are important and yet not wholly mentioned. Would you agree?*

We thank the editor for her advice and mentioned a few resources for scientists that provide assistance in science communication, including the Science Media Centre as well as "
[revised manuscript text omitted]

450